# Occlusion in Bottom Intakes with Circular Bars by Flow with Gravel-Sized Sediment. An Experimental Study

**Juan T. Garcia** [1,*], **Luis G. Castillo** [1], **Patricia L. Haro** [1,2] **and Jose M. Carrillo** [1]

1    Civil Engineering Department, Universidad Politécnica de Cartagena, Paseo Alfonso XIII, 52,
     30203 Cartagena, Spain; luis.castillo@upct.es (L.G.C.); patricia.haro@epn.edu.ec (P.L.H.);
     jose.carrillo@upct.es (J.M.C.)
2    Civil and Environmental Engineering Department, Escuela Politécnica Nacional,
     Ladrón de Guevara E11-253, Quito 170517, Ecuador
*    Correspondence: juan.gbermejo@upct.es; Tel.: +34-968-327-026

**Abstract:** Obstruction of the racks in bottom intakes due to sediment wedged in the slit of the bars can significantly reduce diverted flow. Notwithstanding the design recommendations that are found in the literature, the problem of rack occlusion continues to occur in built structures. This work focuses on the clogging effects in the circular bars of a bottom rack system using gravels whose median diameter, $d_{50}$, is close to the spacing between the bars. An experimental campaign including 24 tests, each repeated time times, with six different longitudinal slopes from 0 to 35% and four different specific incoming flow rates, $q_1$, in the range of 0.115 to 0.198 m$^3$/s/m, is presented. The results show the inefficiency of circular profiles in comparison with T-shaped bars. No important influence of rack slope is found that could help to reduce clogging. This works confirms the importance of the selection of bar profile to reduce maintenance labor. A comparison of results with previous works with gravel sediment in T-shaped bars is considered. A methodology to calculate the wetted rack length considering occlusion due to flow with sediment transport is proposed, and the results are compared with those in the bibliography.

**Keywords:** bottom racks occlusion; circular bars; wetted rack length; effective void ratio

## 1. Introduction

The need to provide water and electricity to populations settled in areas with special geographical, topographic, and hydrological characteristics has led several researchers to focus their attention on the study of bottom intake systems, since stepped torrents (1% < slope < 10%) and laden sediment transport require suitable bottom intake systems for water collection [1]. Small Hydropower Plants (SHP) have been identified as one of the most important energy sources that can provide convenient and uninterrupted energy to remote rural communities or industries in mountain areas, where bottom intake structures are suitable to collect water [2]. In Andean cities which have irregular topography characteristics, the design and construction of bottom intake systems constitute an adequate technical solution for the provision of irrigation water and hydroelectricity. Figure 1 shows a mountain river near rural communities in the Sincholagua Paramo in Ecuador. These communities need a water supply for irrigation purposes, to improve and optimize the production of their fields or as a multipurpose project to enhance tourism.

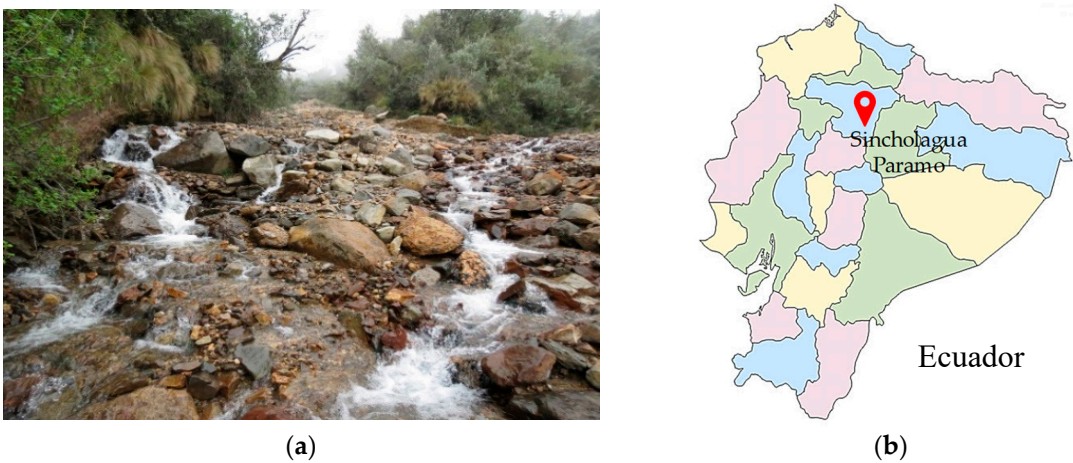

**Figure 1.** Mountain River at Sincholagua Paramo, Ecuador, approximately 4000 meters above sea level, MASL. (**a**) Type of solids transported by the river; (**b**) location in Ecuador (without scale).

Nowadays, in Ecuador, as part of the Mazar–Dudas Hydroelectric Project, San Antonio and Dudas Power Plants are being constructed and both intakes consist of a bottom rack embedded in a weir and designed to derive water at a rate of 4.4 m$^3$/s from the Mazar River and 3.0 m$^3$/s from the Pindilig River [3]. Maintenance works at San Antonio SPH become necessary as parts of the trash rack are obstructed by wedged stones, leaves, or branches, meaning that the collection of the minimum amount of water through the bottom rack can no longer be ensured. This situation is shown in Figure 2.

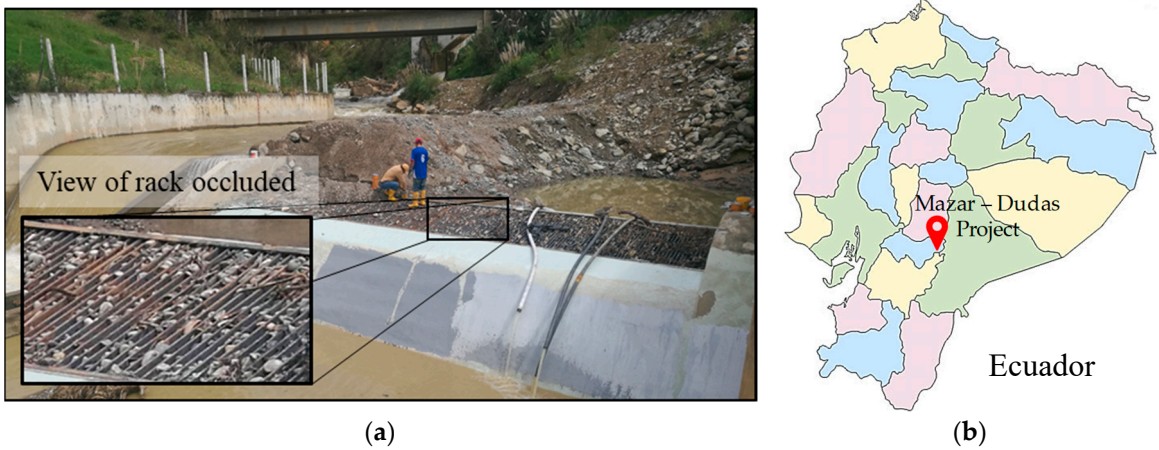

**Figure 2.** Mazar–Dudas Hydroelectric Project: (**a**) San Antonio bottom rack intake embedded in a weir—cleaning tasks and repairing process after a flood in the Mazar River, Ecuador (Image courtesy of Corporación Hidroeléctrica del Ecuador-CELEC EP Hidroazogues) [3]; (**b**) project location in Ecuador (without scale).

To quote another example, in Bolivia there are at least seven irrigation systems with bottom racks of importance. Nevertheless, their design and construction are based on standard models developed for mountain regions and several design issues have been found, such as inadequate intake location, destruction of dissipation structures, rack occlusion, and high costs of intake maintenance. These have caused failures in operation or the partial or total destruction of the bottom racks [4].

Clogging of the racks is considered to be one of the most important causes of malfunction in bottom intakes [1,2,4]. The principal design parameters of the racks are: the space between the bars, $b_1$; their width, $b_w$; the longitudinal slope; and the bar profile adopted. The optimum bar profiles differ when dealing with clear water as compared to when transported sediments are included. In the latter situation, clogging needs to be considered in order to minimize maintenance and operation labor.

This has been stated by several authors [1,2,5–7]. The optimum bar profiles are presented in Figure 3. This figure presents the optimum profiles for, on the one hand, maximizing derived flow without considering sediments (Figure 3a) [5,8,9], and on the other, minimizing the sediment trapped in the slits of the bars (Figure 3b) as suggested in Reference [2] and the present work. It can be observed that some of the optimum profiles in the case of clear water diversion are less efficient in the case of sediment transport. This fact is related to the length of contact between embedded gravels and profiles. Figure 3c presents how rounded profiles with a higher radius, such as circular bars, present a larger length of gravel–bar contact than other crest-rounded bars with a lower radius, which are more efficient. The length of gravel–bar contact is proportional to the drag force needed to remove gravels embedded in the slit between two bars. Conversely, Brunella et al. [10] conducted an experimental work for clear water diversion with circular profiles and recommended those to avoid clogging. This contrasts with other authors' experiences [2,6]. No experimental measurements were found related to circular bar profiles in bottom intakes considering sediment transport.

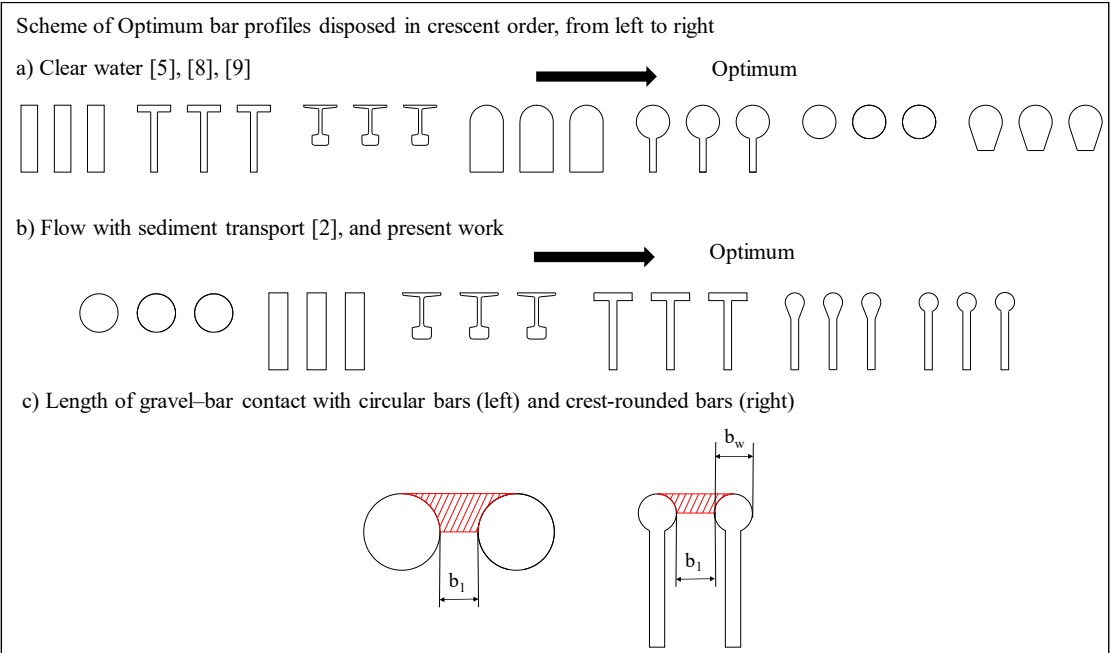

**Figure 3.** Scheme of optimum bar profiles: (**a**) clear water flow; (**b**) flow with sediments; (**c**) comprehensive sketch to visualize the influence of the profile radius on the length of gravel–bar contact.

From experimental works in the laboratory and based on past experience in the maintenance and operation of these structures, several authors proposed design parameters to reduce clogging. Table 1 summarizes recommendations in the bibliography. These recommendations address the increasing of: bar space, $b_1$; longitudinal rack slope, *tanθ*; or rack length to take into account clogging and/or the consideration of an obstruction factor. As an example, the equation proposed by Krochin [6] to calculate the wetted rack length to derive an incoming flow includes an obstruction coefficient to take into account occlusion in the bottom intake design:

$$L = \left[ \frac{0.313q_1}{(C_q k)^{3/2}} \right]^{2/3},$$

(1)

where $L$ is the wetted rack length to derive a flow, taking into account sediment transport; $q_1$ is the incoming flow; $C_q$ is the discharge coefficient; $k$ is the obstruction parameter defined as $k = (1-f)m$; $m$ is the void ratio, which is calculated as the void area divided by the total area of the rack; and $f$ is the

percentage of rack that is considered to be occluded (Krochin recommended adopting a value between 0.15 and 0.30).

**Table 1.** Design parameters adopted in bottom intakes considering clogging.

| Author | Bar Space, $b_1$ (m) | Longitudinal Rack Slope (%) | Increment of Rack Length (%) | Obstruction Factor (%) | Bar Profile |
|---|---|---|---|---|---|
| Ract-Madoux et al. [11] | 0.100 | 20 | – | – | Thick trapezoidal, rail-type, round head (next to circular) |
| White et al. [12] | 0.030–0.076 | 20 | – | – | Prismatic heptagon |
| Krochin [6] | 0.020–0.060 | 20 | – | 0.15–0.30 | Prismatic |
| Simmler [13], Drobir [5] | 0.150 ($d_{95} = 0.060$) | 20–30 | 0.50–1.00 | – | Several rounded profiles (next to circular) |
| Lauterjung and Schmidt [1] | – | 9–70 | 0.20 | – | Same as Reference [13] |
| Bouvard [14] | 0.100–0.120/0.002–0.03 SHP | 30–60 | 0.50–1.00 | – | Same as Reference [11] |
| Raudkivi [15] | >0.005 | 20 | – | – | Trapezoidal, inverted railway tracks |
| Andaroodi and Schleiss [2] | 0.020–0.040 SHP | 84–100 | 0.20 | – | Bulb-ended, round head |
| Castillo et al. [7], Carrillo et al. [16], Castillo et al. [17], García [18] | 0.006–0.045 | 30 | – | 0.30 | T-shaped |

Note: $d_{95}$ is the diameter where 95% percent of the distribution has smaller particle size.

In the design of bottom intakes, two different approaches may be distinguished: (a) reduced bar spacing, to prevent the entrance of gravel or; (b) wider bar spacing, which only protects against the diversion of coarse parts of the sediment. In the second case, the sediment is separated in a sand trap afterwards and only the dimensions which may cause problems are restrained by the rack. If the amount of sediment is large, more sediment has to be flushed out and, in this case, more water may be lost. Experimental research work based on the study of the occlusion phenomenon in bottom racks becomes important to optimize the structure sizing, guarantee design flow derivation, and increase the average life span of structures. There are few experimental works the consider trapped sediments available in the bibliography. Longitudinal prismatic bars [19], mesh panels made of prismatic bars [20], and longitudinal T-shaped bars [7] have previously been tested to study the influence of sediment trapped in a diverted flow.

The present work collects the results of 24 tests, wherein each test was repeated three times, in a bottom rack with circular bars with a void ratio (the area between bars divided by the total area) of $m = 0.28$, using gravel with a characteristic diameter, $d_{50}$, of 22.0 mm, and with four different incoming flows and six longitudinal rack slopes. The analysis of clogging effects in the diverted flow and a comparison of the results with those presented previously using T-shaped bars by Castillo et al. [7] are included. The methodology to calculate the length of rack needed to consider the clogging phenomenon is also included and compared with literature recommendations such as those of Krochin [6] and Drobir [5].

## 2. Experimental Setting

### 2.1. Physical Device

The physical device consists of an intake system based on Noseda's [21] physical model (Figure 4). The inlet is a 5.00-m-long and 0.50-m-wide channel with methacrylate walls. At the end of the channel, there is a bottom rack intake system with different slopes (from horizontal to 35%). The racks were built with aluminum bars with circular profiles. The rack length is 0.90 m in the flow direction. Bars are longitudinally oriented with the flow direction. Table 2 presents the characteristics of the racks used in

the present work, as well as those used in the previous works [7] to provide a comparison with the present results.

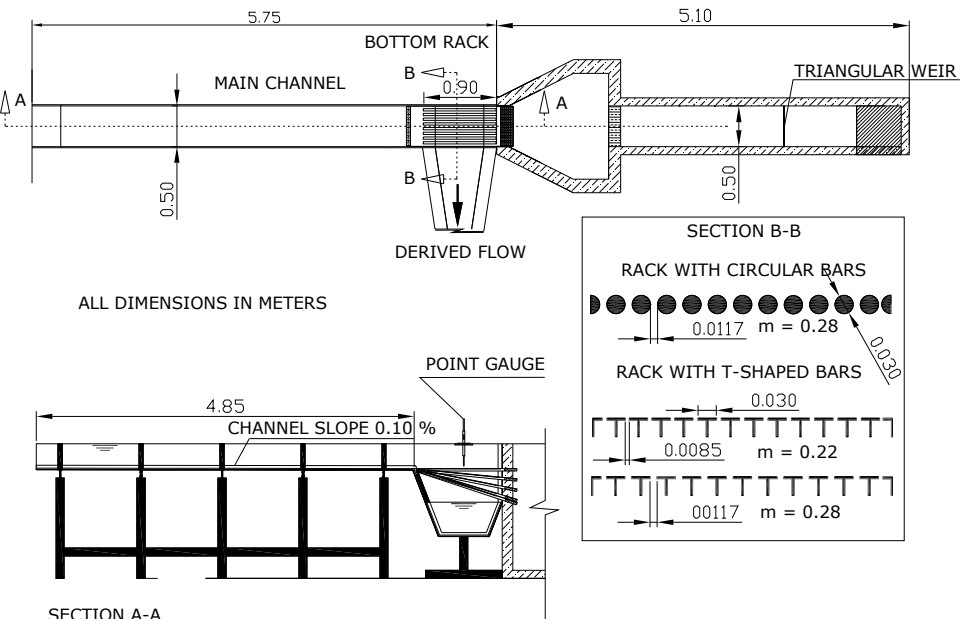

**Figure 4.** Scheme of the intake system physical device.

**Table 2.** Characteristics of the racks tested with gravels in the present and previous works.

| Description | Rack Length (m) | Rack Width (m) | Bar Type (mm) | Width of the Bars, $b_w$ (mm) | Direction of the Bars | Spacing between Bars, $b_1$ (mm) | Void Ratio $m = \frac{b_1}{b_1 + b_w}$ | Longitudinal Slope (%) |
|---|---|---|---|---|---|---|---|---|
| Present work | 0.90 | 0.50 | O30/30 | 30 | Longitudinal | 11.7 | 0.28 | 0, 10, 20, 30, 33, and 35 |
| Previous works [7] | 0.90 | 0.50 | T30/25/2 T30/25/2 | 30 | Longitudinal | 8.5 11.7 | 0.22 0.28 | 0, 10, 20, 30, and 33 |

Inlet specific flows were in the range of 0.115 to 0.198 m$^3$/s/m to assure that the gravels were transported by the flow. The inlet total flow was measured with an Promag 53-W electromagnetic flowmeter of 125 mm (Endress Häuser, Reinach, Basel, Swizterland) with an accuracy of 0.5%. Tests were performed with six different longitudinal rack slopes. In all cases, the approximation flow regime was subcritical, changing to supercritical near the bottom racks. Further details of the model can be found in References [7,16–18,22]. The rejected flow was measured by a 90-degree V-notch weir.

## 2.2. Sediment Experimental Tests with Racks Made of Circular Bars (m = 0.28)

Occlusion phenomena of circular bottom racks were evaluated using gravel with a median grain size of $d_{50}$ = 22.0 mm (the sieve curve is almost uniform) and which presented rounded faces. At the coarse part of the sieve curve, $d_{90}$ = 35 mm and $d_{max}$ = 40 mm. At the finest part, $d_{min}$ = 10 mm, while $d_{10}$ = 16 mm. Considering the spacing between bars, $b_1$ = 11.7 mm, few materials will go through the rack for the tested gravel. Zingg's shape classification [18] for this gravel is presented in Table 3.

**Table 3.** Zingg's shape classification.

| $d_{50}$ (mm) | Blade | Disc | Rod | Sphere |
|---|---|---|---|---|
| 22.0 | 8% | 30% | 19% | 43% |

Source: Reference [18].

Four specific inlet flows ($q_1$ = 114.6, 138.3, 155.5, and 198.0 l/s/m) and six rack slopes (*tanθ* = 0%, 10%, 20%, 30%, 33%, and 35%) were tested. Twenty-four tests, with each test repeated three times, were conducted in the laboratory.

A 100-kilogram gravel bed was placed along the approach channel. The flow dragged the solids towards the bottom rack, then passing over the bottom racks. One part was diverted passing through the slits between the bars, another was retained between the bars—occluding the space between the bars—and the rest continued downstream. After all the sediment had passed over the rack or was deposed in the spacing between bars, the test duration was extended until no movement of the gravels was observed. During the dosage of gravels, an equilibrium between the supply and transport was achieved to prevent all of the gravels from passing through the rack at the same time. In case of the lower incoming flows, $q_1$ = 114.6 and 138.3 l/s/m, the duration of the test was approximately 12 min, whilst for the high flow rates the duration of the test was approximately 8 min. At the end of each test, the solid weight of each of the three parts (derived, rejected, and trapped sediment) was quantified. Gravels were drained for a period of 30 min before being weighed.

### 2.3. Previous Studies of T-Shaped Bottom Rack Occlusion by Flow with Gravel-Sized Sediment

Previous works conducted by Castillo et al. [7] presented 90 tests conducted using three distinct types of gravel with the following values of $d_{50}$: 8.3 mm (gravel 1), 14.8 mm (gravel 2), and 22.0 mm (gravel 3). The first type of gravel was tested with a T-shaped flat rack with a void ratio of $m$ = 0.22, while the two remaining types of gravel were used for the T-shaped flat bar with $m$ = 0.28. They were tested with three incoming flows (114.6, 138.3, and 155.5 l/s/m) and five different slopes (0%, 10%, 20%, 30%, and 33%). Further details about that study can be obtained in References [7,18]. As a result of the experiments in the laboratory, the following was obtained:

- Reduced void ratio according to rack occlusion, termed the effective void ratio, $m'$.
- Visualization of preferential occlusion area related to the streamline curvature.
- The most efficient longitudinal rack slope, which in T-shaped bars was 30%.
- Finally, a methodology was proposed to obtain the wetted rack length, taking into account the sediment transport and occlusion as well as its comparison with the lengths proposed by Krochin [6].

The results obtained in the present work are compared with those of Castillo et al. [7] with T-shaped bars. The information compared corresponds to the same void ratio of the rack, i.e., $m$ = 0.28 and gravels (2) and (3), with $d_{50}$ = 14.8 mm and 22.0 mm, respectively.

### 2.4. Methodology to Define the Effective Void Ratio

Gravels embedded in the slit between two bars give rise to a reduction in the open area of the rack, which results in a lower void ratio. This reduced void ratio, $m'$, is termed the effective void ratio. To define the effective void ratio, $m'$, it is proposed to numerically solve Equations (2) and (3) to obtain the flow depth and the derived flow along the rack. Equation (2) is obtained through coupling the orifice equation and the derivative of the energy equation as collected in References [16,17]. Equation (3) was first presented in previous experimental works of Carrillo et al. [16] for bottom racks with circular bars and the same void ratio. Equations (2) and (3) are solved by a fourth-order Runge–Kutta algorithm with a trial and error process varying the value of the effective void ratio, $m'$, until the rejected flow calculated at the end of the rack agrees with the value measured in the laboratory.

$$\frac{dh}{dx} = \frac{2m'C_{qH}\sqrt{(H_0 + x\sin\theta)(H_0 + x\sin\theta - h\cos\theta)} + h\sin\theta}{3h\cos\theta - 2(H_0 + x\sin\theta)}, \tag{2}$$

$$C_{qH} \approx \frac{1 - 0.15\left[1 - 0.51(1 - 0.45tg\theta)\left(\frac{x}{h_c}m'\right)\right]^{-2.7}}{(1 + tg\theta)}, \tag{3}$$

where $h$ is the flow depth; $x$ is the longitudinal coordinate; $C_{qH}$ is the discharge coefficient; $H_0$ is the energy at the beginning of the rack considered equal to the minimum energy; $\theta$ is the angle of the rack with horizontal; and $h_c$ is the critical depth (Figure 5). The numerical computation interval for $x$ is 0.05 m, until it reaches the total rack length of 0.90 m.

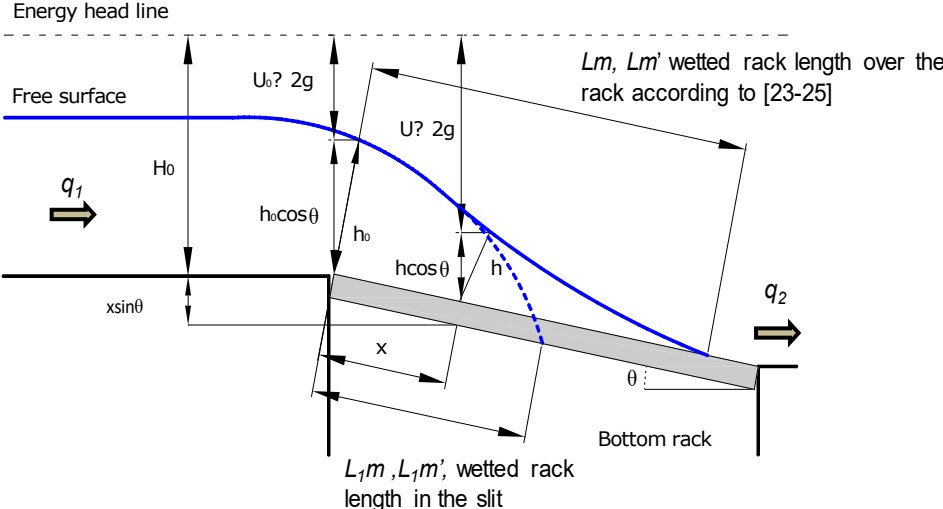

**Figure 5.** Scheme of the spatially varied flow over the bottom intake system.

Figure 5 also shows the lengths for the design of the bottom racks [23–25], which are as follows: the wetted rack length to completely derive an incoming flow with original void ratio, $Lm$; the effective wetted rack length considering rack occlusion, $Lm'$; the wetted rack length in the slit of two bars, considering the initial void ratio, $L_1m$; and the wetted rack length in the slit of two bars, considering the effective void ratio, $L_1m'$. $m'$ denotes the effective void ratio that takes into account the reduction of the original void ratio due to the occlusion of the rack as a result of gravel deposition.

## 3. Results and Discussion

### 3.1. Sediment Tests with Circular Bars

#### 3.1.1. Deposition of Gravels over Racks

During the tests, part of the gravels transported with the flow was trapped in the space between the bars, producing a partial occlusion of the bottom rack. This supposes a reduction in the efficiency of the derived flow in comparison with the clear water flow case. The rejected flow was measured with the 90-degree V-notch weir for each of the incoming flows studied. Results of the efficiency of the derived flow are presented in Figure 6. This figure shows the difference between incoming and rejected flows for each longitudinal slope divided by the incoming flow, $(q_1 - q_2)/q_1$, which is the percentage of the derived flow. The values for the clear water case are also included. In Figure 6, it can be observed that the increase in the slope does not suppose a remarkable increase in the efficiency, as was expected in view of the results of other racks, such as those with T-shaped bars [7]. In the case of racks made with circular bars, the adopted form between two bars enabled the possibility that gravels of sizes even larger than the space between bars can be embedded (Figure 3c). Moreover, the surface of the gravel–bar contact reached when gravel was trapped was greater than that in other bar types. This means that the drag force exerted by the water to prevent the deposition of gravel was not high, enough even with large longitudinal slopes such as 35%. The difference between the percentage of derived flow from the lowest and the highest slope adopted in the laboratory facility was in a range of 6% for $q_1 = 138.3$ l/s/m, and 11% for $q_1 = 198.0$ l/s/m. These values are also summarized in Table 4,

which presents the mean value of the three tests for each case. In this table, the efficiency is also shown to decrease with the increase in the incoming flow as the rack has a constant length of 0.90 m.

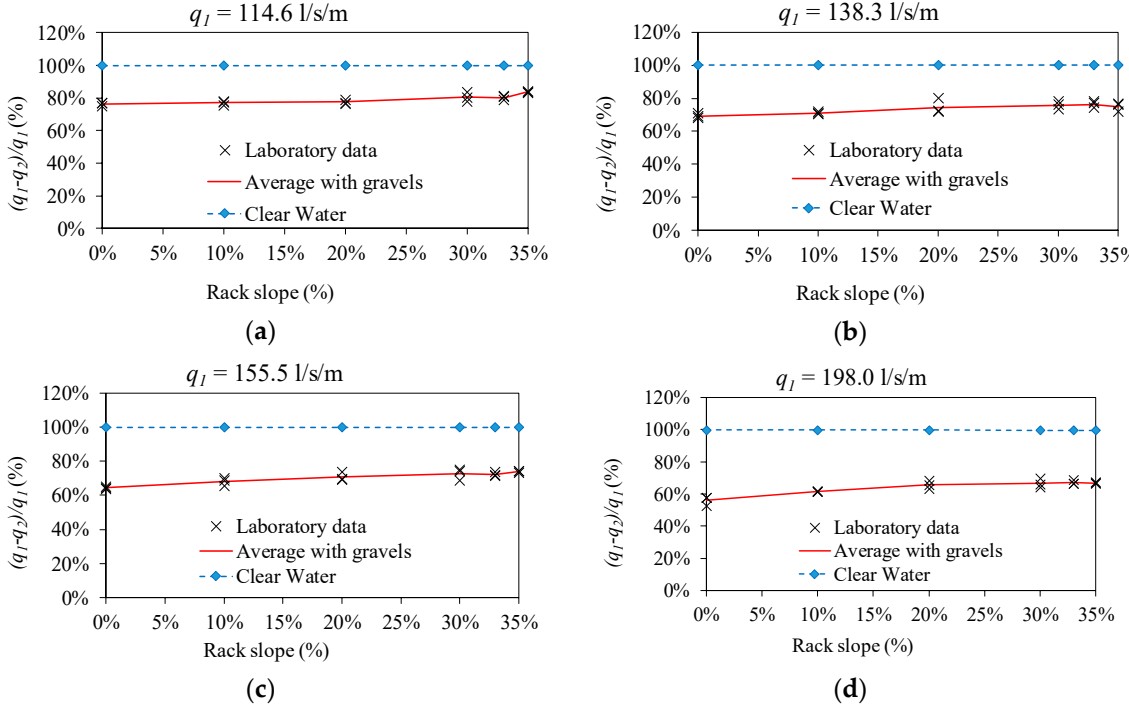

**Figure 6.** Percentage of derived flow for a specific inlet flow: (**a**) 114.6 l/s/m; (**b**) 138.3 l/s/m; (**c**) 155.5 l/s/m; (**d**) 198.0 l/s/m.

**Table 4.** Percentage of derived flow.

| Incoming Flow (l/s/m) | Percentage of Derived Flow (%) | | | | | |
|---|---|---|---|---|---|---|
| | Longitudinal Slope (%) | | | | | |
| | 0 | 10 | 20 | 30 | 33 | 35 |
| 114.6 | 76 | 77 | 78 | 81 | 80 | 84 |
| 138.3 | 69 | 71 | 74 | 76 | 76 | 75 |
| 155.5 | 64 | 68 | 71 | 73 | 72 | 74 |
| 198.0 | 56 | 62 | 66 | 67 | 67 | 67 |

### 3.1.2. Effective Void Ratio

Initially, this reduced void ratio was calculated by subtracting the area occupied by gravels from the initial area. Photographs of the top view of the occluded rack were taken at the end of each test to be processed later with a software in CAD design (AutoCAD 2016 20.1). First, the photographs were imported and the size was adjusted to their real dimensions. Then, a line was drawn over the gravels deposited along each of the 12 slits of each rack. This line (in red) can be observed in Figures 7 and 8. Once the line was drawn, its length was divided by the original length of the rack, thus giving the percentage of reduction of the original void ratio. The next step was to calculate the average reduction between all the slits of the rack, finally giving the effective void ratio. This was repeated for each of the tests taken. Hence, the initial and final void ratios could be compared. It was observed that low rack slopes presented the highest deposition of gravels, in accordance with other works [7].

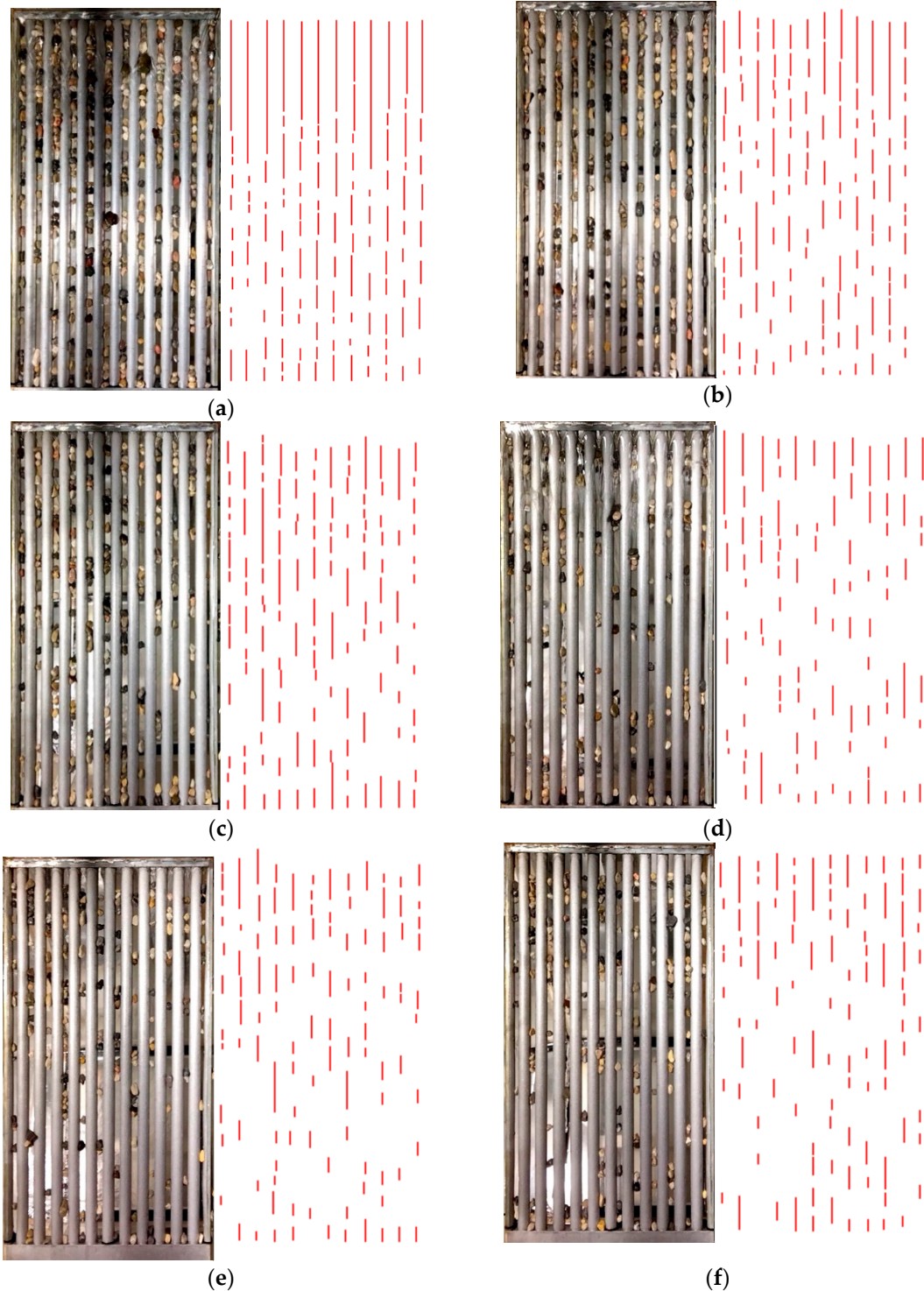

**Figure 7.** Top view of occluded bottom racks with circular bars for $m = 0.28$, $q_1 = 138.3$ l/s/m, and diverse longitudinal slopes: (**a**) 0%; (**b**) 10%; (**c**) 20%; (**d**) 30%; (**e**) 33%; and (**f**) 35%.

In Figure 8, the void ratio can be observed, as defined through top images of the racks at the end of each test, dividing each rack into four parts. It can also be observed that the occlusion is higher at the beginning of the rack and that the increment of the incoming flow and the slope reduce the occluded areas.

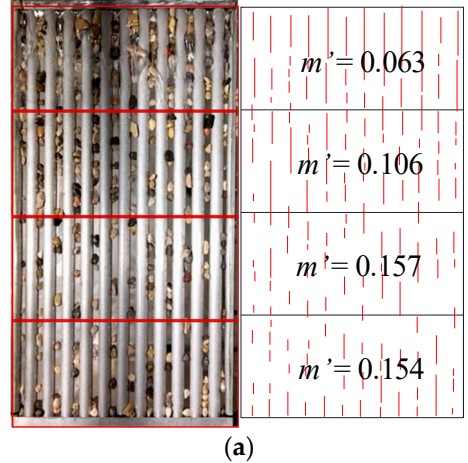
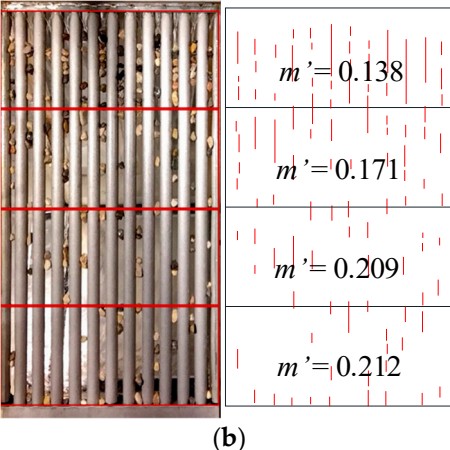

**Figure 8.** Top view of occluded bottom racks with circular bars for *m* = 0.60, dividing the rack into four parts, with the void ratio calculated from occluded areas: (**a**) $q_1$ = 155.5 l/s/m and 10% longitudinal slope and (**b**) $q_1$ = 198.0 l/s/m and 35% longitudinal slope.

The wake formation when flow passes around gravels could not be easily considered when determining the effective void ratio using photographs. Figure 9 shows how the flow around gravels generates wakes which amplify the effect of the occluded area even more. Thus, to define the effective void ratio, $m'$, it is proposed to numerically solve Equations (2) and (3) to obtain the flow depth and the derived flow along the rack, as proposed in Section 2.4.

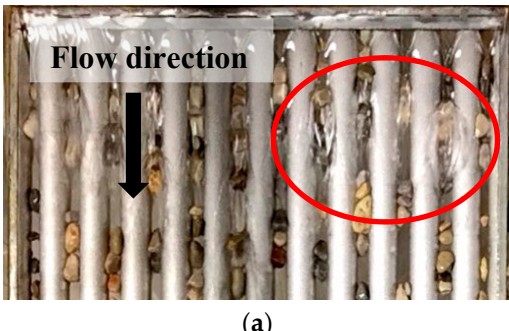
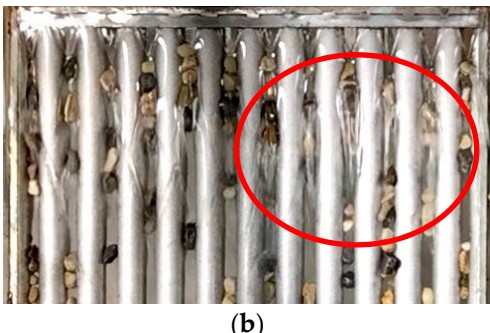

**Figure 9.** Flow over the occluded circular bars bottom rack and formation of wakes for rack slopes: (**a**) 20%; (**b**) 30%.

The results obtained solving Equations (2) and (3) with flow values passing along the rack were in agreement with those measured in the laboratory and are shown in Figure 10. The average values calculated varies from $m' \approx 0.071$ ($tan\theta$ = 0%) to $m' \approx 0.083$ ($tan\theta$ = 35%) when the initial void ratio was *m* = 0.28. It was observed that an important reduction of the void ratio occurred due to the embedment of gravel in the slits of the bars. This remained almost constant for all the incoming flows and longitudinal slopes adopted. The influence of the longitudinal slope of the rack seemed to be limited and it was not as important as expected in comparison with other bar types. The reduction of the void ratio was around two-thirds of the initial void ratio, as can be observed in Figure 10. Table 5 presents the effective void ratios calculated for each case.

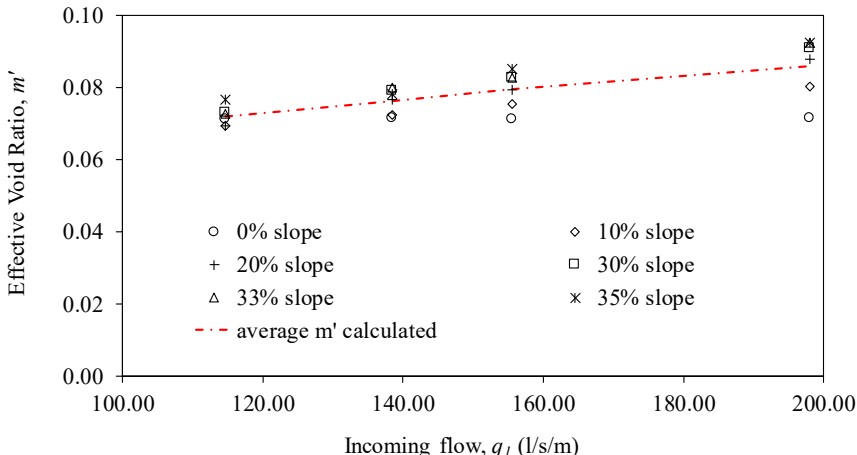

**Figure 10.** Effective void ratio values, $m'$, for different rack slopes and incoming flows.

**Table 5.** Effective void ratios from Equations (1) and (2).

| Incoming Flow (l/s/m) | Effective Void Ratio | | | | | |
|---|---|---|---|---|---|---|
| | Longitudinal Slope (%) | | | | | |
| | **0** | **10** | **20** | **30** | **33** | **35** |
| 114.6 | 0.070 | 0.069 | 0.069 | 0.073 | 0.073 | 0.077 |
| 138.3 | 0.071 | 0.073 | 0.077 | 0.079 | 0.080 | 0.078 |
| 155.5 | 0.071 | 0.075 | 0.079 | 0.083 | 0.083 | 0.085 |
| 198.0 | 0.072 | 0.080 | 0.088 | 0.091 | 0.092 | 0.092 |

The occlusion of the space between the bars reduced the efficiency of the derived flow and resulted in an increment of the flow depth along the rack. Once the effective void ratio was calculated from using Equations (2) and (3), the flow profile was also obtained considering the void ratio reduction. Figures 11–13 present the flow profile with clear water measured in the lab and collected in previous works [16] as well as the calculated flow profile with the reduced void ratio for the incoming flows of 138.3, 155.5, and 198.8 l/s/m and the slopes from horizontal to 30%. The flow profile calculated with the void fraction reduced was in agreement with that observed in the lab when compared with the flow profile observed in the photographs. In these figures, when comparing the surface flow profile with clear water and that with a sediment-laden flow that produces obstructions in the slits of the racks, an important difference in the flow depth and an increment in the wetted rack length are observed, exceeding the available rack length of 0.90 m in the lab in all cases. As the inlet flow is supercritical, the first section of the rack shows that is it slightly affected by the rack occlusion.

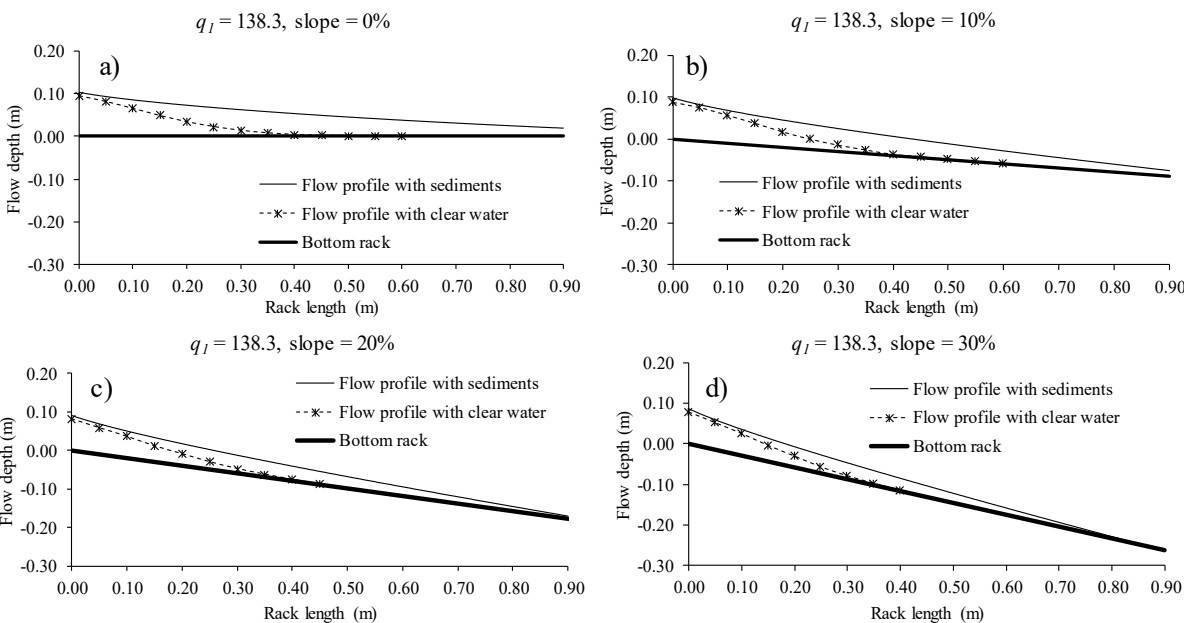

**Figure 11.** Water profiles over the rack measured in the laboratory and calculated for $q_1 = 138.3 \, \text{l/s/m}$ and slopes (**a**) 0%; (**b**) 10%; (**c**) 20%; and (**d**) 30%.

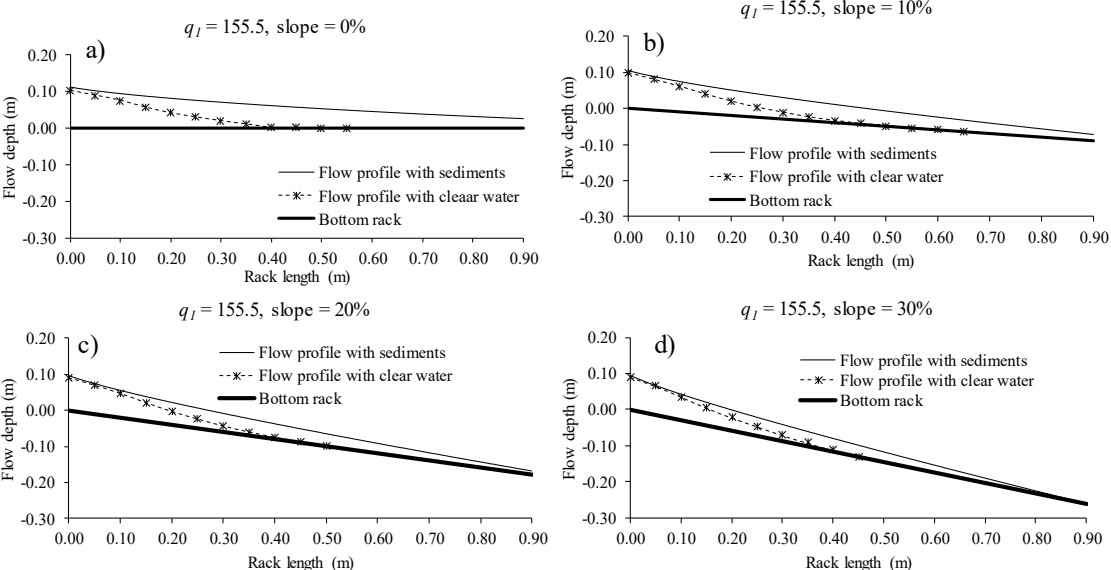

**Figure 12.** Water profiles over the rack measured in the laboratory and calculated for $q_1 = 155.5 \, \text{l/s/m}$ and slopes (**a**) 0%; (**b**) 10%; (**c**) 20%; and (**d**) 30%.

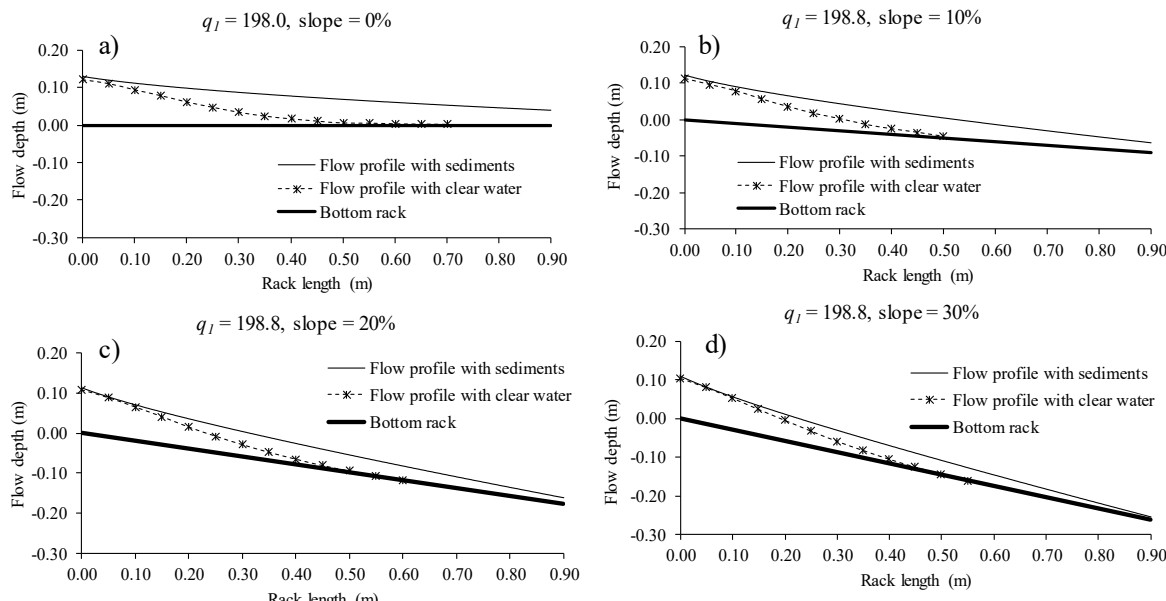

**Figure 13.** Water profiles over the rack measured in the laboratory and calculated for $q_1 = 198.0\,\text{l/s/m}$ and slopes (**a**) 0%; (**b**) 10%; (**c**) 20%; and (**d**) 30%.

### 3.2. Relation between Hydraulic Parameters at the Beginning of the Rack with Ratio m'/m

The correlation between hydraulic parameters with the effective void ratio, $m'/m$, measured in the lab is of interest for each incoming flow and longitudinal slope. Finding this relationship, the occlusion conditions could be obtained from hydraulic parameters that are more easily quantified.

Regarding the obstruction of the racks, the resultant force exerted by the fluid on the embedded gravels depends upon the geometrical form of the body, the roughness of its surface, its relative velocity, and the fluid density and viscosity. For a given geometrical form and roughness, the Buckingham $\Pi$ theorem provides two dimensionless parameters, the Reynolds number and a dimensionless parameter related to the drag force. These variables together with the velocity, all measured at the beginning of the rack, are analyzed and presented in following figures, in relation to the ratio between the effective void ratio after obstruction divided by the original void ratio, $m'/m$. These variables are defined as:

$$R_{e0} = \frac{U_0 d_{50}}{\nu} \tag{4}$$

$$F_{D0} = \frac{1}{2}\rho U_0^2 C_D \pi (0.5 d_{50})^2 \tag{5}$$

where $U_0$ is the velocity at the beginning of the rack; $d_{50}$ the median diameter of the gravels; $\nu$ the kinematic viscosity of the water; $\rho$ the density of the water; and $C_D$ is the drag coefficient of the gravel that adopts the value of 0.45.

These variables are presented in Figure 14 for all the cases measured in the lab. It can be observed that linear adjustments between $m'/m$ ratio and the variables were achieved. Although a better correlation was found in case of the drag force, $F_{D0}$, the velocity was proposed to be used to calculate the $m'/m$ ratio, as the velocity can be obtained in an easier way in practice. Garcia et al. [23] proposed a relation between the incoming flow and the initial flow depth, $h_0$. Once the velocity at the beginning of the rack can be calculated from the incoming flow, the reduction of the void ratio can be obtained, which supposes a useful relation that can be used in the design of bottom intake systems.

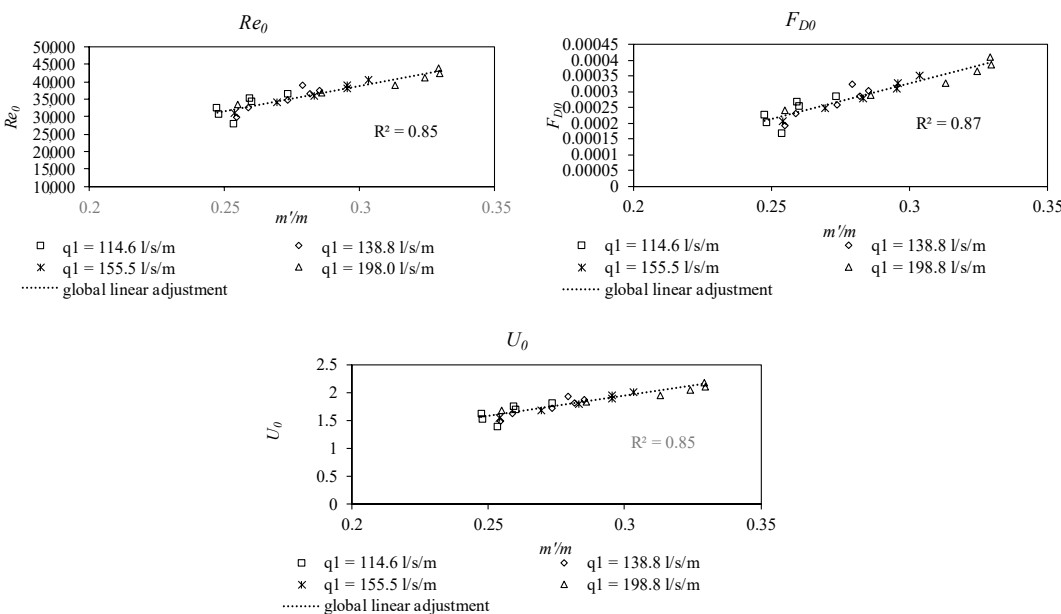

**Figure 14.** Velocity at the beginning of the rack, $U_0$, as a function of the ratio between the effective and the initial void ratios, $m'/m$.

To calculate the velocity at the beginning of the rack, $U_0$, we used measurements previously reported in Garcia et al. [23], wherein the relation $h_0 = a\left(q^{2/3}\right)$ was achieved for each longitudinal slope. Taking into account that $h_c = \left(q/\sqrt{g}\right)^{2/3}$, the previous relation can be presented as $\frac{h_0}{h_c} = a\left(\sqrt{g}^{2/3}\right)$. Values of $\frac{h_0}{h_c}$ are presented in the case of circular bars and the void ratio $m = 0.28$ is employed for several longitudinal slopes. A linear regression is proposed in Figure 15 and collected in Equation (4) that relates $\frac{h_0}{h_c}$ to the longitudinal rack slope, with a correlation coefficient of $R^2 = 0.85$. Once $\frac{h_0}{h_c}$ is calculated, we can obtain $U_0$ from the equation $U_0 = q/\left[\left(q/\sqrt{g}\right)^{2/3}\left(\frac{h_0}{h_c}\right)_{regression}\right]$.

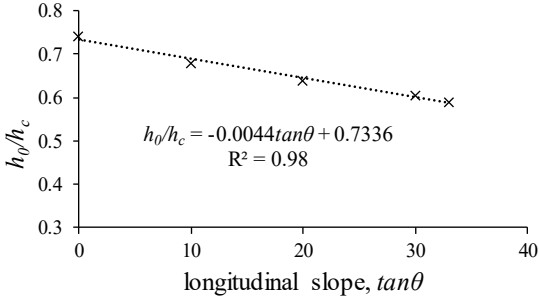

**Figure 15.** Ratio $h_0/h_c$, which is dependent on the rack slope, according to Reference [23].

### 3.3. Comparison of Occlusion in Racks with Circular and T-Shaped Flat Bars

In previous works, Castillo et al. [7] carried out experimental campaigns with gravel-sized sediments and T-shaped flat bars (Figure 4). A comparison of the results obtained in the present work with those obtained for T-shaped flat bars is presented in Figures 16–18 for the incoming flows of 114.6, 138.3, and 155.5 l/s/m. The characteristics of the gravels used in previous works were collected in Section 2.3. The gravels had a median grain size of $d_{50} = 14.80$ mm with fractured faces material (named gravel 2) or a grain size of $d_{50} = 22.00$ mm with rounded faces (named gravel 3). In the present work, the same gravel with $d_{50} = 22.00$ mm was used. Efficiencies are presented in Figure 16, and it can be seen that the highest percentage of derived flow in each case corresponds to the tests carried out with the T-shaped bars, while the lowest values are presented with the circular bar test. It can be observed

that in terms of flow derivation capacity between the racks with circular bars and those with T-shaped bars, the circular bars presented the lowest efficiency. The values presented in Castillo et al. [7] were reviewed in the case of gravel with a grain size of $d_{50}$ = 14.80 mm.

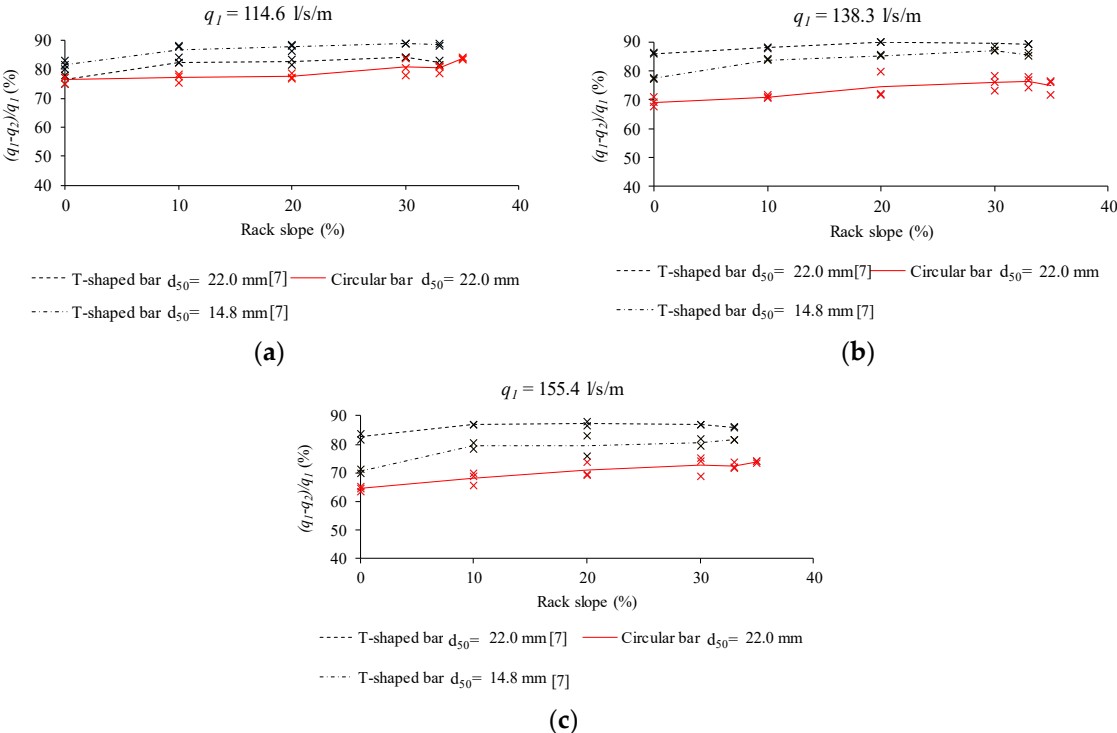

**Figure 16.** Comparison between percentage of derived flow for bottom racks with T-shaped bars (gravel 2 and gravel 3) and circular bars (gravel 3) for specific inlet flows: (**a**) 114.6 l/s/m; (**b**) 138.3 l/s/m; and (**c**) 155.5 l/s/m.

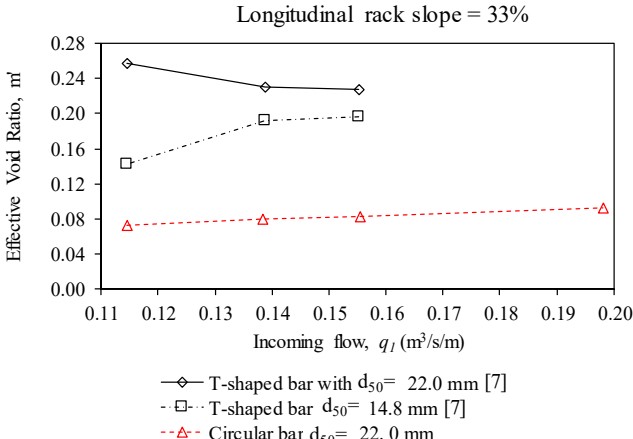

**Figure 17.** Comparison of effective void ratio, $m'$, for circular bars and values previously presented for T-shaped bars [7]. All cases have a void ratio of $m$ = 0.28.

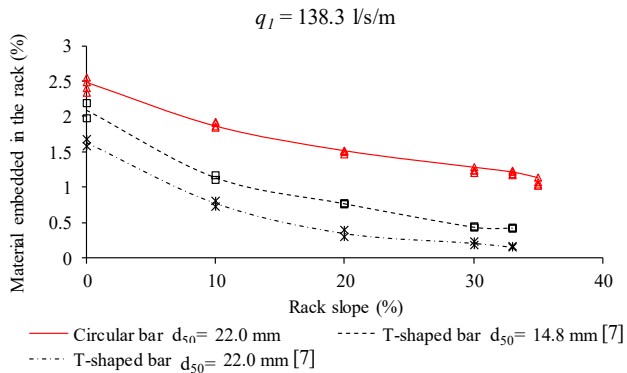

**Figure 18.** Comparison of effective void ratio, $m'$, for circular bars and values previously presented for T-shaped bars [7].

Figure 17 compares the effective void ratio, $m'$, calculated in the case of circular bars with values previously presented for T-shaped bars [7] and the same gravel-sized sediments. It can be observed that the T-shaped bars continued to exhibit a notably higher value of $m'$ in comparison with the circular bars when the flow with gravel traveled through the rack. The T-shaped bars also reached higher values of $m'$ when the incoming flow was increased, whilst the circular bars presented minor variations with the incoming flow. As already commented, this is due to the fact that the sediments embedded in the circular bars were of high weight and the gravel–bar contact length was quite large, thus requiring important drag forces to prevent the sediments from becoming trapped. The quantity of materials, in percentage of incoming weight, trapped in the slits of the bars for both circular and T-shaped bars is presented for the case of an incoming flow of $q_1 = 138.3$ l/s/m in Figure 18. It can be observed that this quantity was higher in the case of circular bars, which makes maintenance more difficult.

*3.4. Methodology to Calculate the Effective Wetted Rack Length, Lm', for the Design of Bottom Intakes Considering the Gravel-Sized Sediments*

A methodology is proposed in this section to calculate the length of the rack necessary to completely derive an incoming flow, taking into account the influence of the gravels, $Lm'$. To calculate this effective wetted rack length, a model was first proposed to obtain the effective void ratio, $m'$, for any longitudinal slope and incoming flow; the statistical model proposed by Castillo et al. [7] was considered and extended for the racks made by circular bars, as presented in Figure 19, including previous results for the case of T-shaped bars. The model consists in a linear regression presented in Equations (4) and (5). The model requires the velocity at the beginning of the rack, $U_0$, which can be calculated as explained in Section 3.2.

$$\text{Circular bars}: \frac{U_0(d_{50c}/b_1)}{W^{0.205}} = 28.90m'/m - 0.92 \tag{6}$$

$$\text{T-shaped bars [7]}: \frac{U_0(d_{50c}/b_1)}{W^{0.205}} = 3.07m'/m + 2.41 \tag{7}$$

where $d_{50c}$ is the diameter that corresponds to the smallest of the three axes of the ellipsoid, which represents 50% of the weight of the gravels embedded in the rack at the end of each test; $W$ is the mean weight of the materials embedded in the rack. The values of these parameters in the case of circular bars were $d_{50c} = 19.5$ mm and $W = 10.2$ g. The correlation coefficients of Equations (6) and (7) were 0.85 and 0.71, respectively.

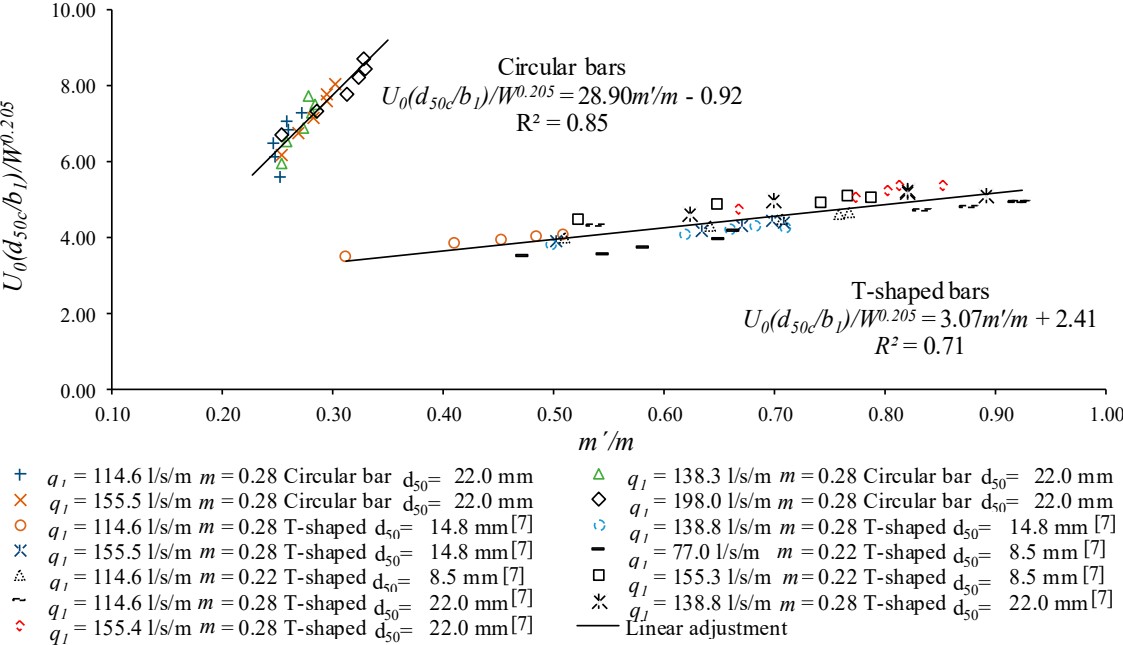

**Figure 19.** Linear adjustment of the ratio $m'/m$ ratio as a function of the values $U_0$, $d_{50c}/b_1$, and $W$.

Once the effective void ratio was calculated, the next step was to calculate the wetted rack length. As stated in previous works by Garcia et al. [23], in order to derive a certain flow in the wetted rack length, the length where flux sticks to the bar due to surface tension must be taken into account [23–25]. Therefore, to calculate the wetted length able to derive a certain flow with sediment transport, the following steps were proposed:

i.  Calculate $L_1m$, the wetted rack length in the slit of two bars, considering the initial void ratio, $m$, by using Equation (2) coupled with Equation (3) in the case of circular bars or with Equation (8) in the case of T-shaped bars [17]:

$$C_{qH} \approx \frac{0.58e^{-0.75(\frac{x}{h_c}m)}}{(1+0.9tg\theta)};$$ (8)

ii. Calculate the effective void ratio from the proposed Equations (6) and (7) depending on the rack slope and velocity at the beginning of the rack, $U_0$;

iii. Calculate $L_1m'$, the wetted rack length in the slit of two bars, considering the effective void ratio that takes into account the clogging effects obtained, $m$, by using Equation (2) coupled with Equation (3) in the case of circular bars, or with Equation (8) in the case of T-shaped bars;

iv. Calculate $Lm$, i.e., the wetted rack length over a bar using the methodology of Garcia et al. [23].

$$Lm = \frac{q}{C_{qH}m\sqrt{2gH_{min}}},$$ (9)

$$\overline{C_{qH}} = \frac{amC_{q0}}{(1+tan\theta)}q^b,$$ (10)

where $q$ is the flow derived by the rack; $Lm$ is the wetted rack length; $m$ is the void ratio, $\overline{C_{qH}}$ is the mean discharge coefficient for each wetted rack length; $g$ is the gravitational acceleration; and $H_{min}$ is the minimum energy head calculated as $1.5 h_c$, with $h_c$ being the critical depth in reference to the plane of the rack. Constants $a$ and $b$ in the case of the void ratio $m = 0.28$ adopted the values 1.45 and 0.05, respectively, in the case of circular bars, and 1.50 and 0.05 in the case of T-shaped bars [23]. Figure 5 presents the scheme of the different lengths described.

Figure 20 shows the wetted rack lengths calculated, taking into account gravel-sized sediments for the case of 30% rack slope and both T-shaped and circular bars. The case of T-shaped bars and 10% slope is also included. The incoming specific inlet flows adopted varied from 100 l/s/m to 500 l/s/m, according to the flow rates usually observed in the Mazar–Dudas SPH project and Bolivian bottom intakes [3,4]. The lengths were compared with those proposed by Frank [8,9], which are nowadays present in several hydraulic manuals [1,2,5]. The lengths proposed by Krochin [6] are also included in the figure for an occlusion factor $f$ of 15% and 30% (values of $L/h_c$ of 11.30 and 9.5, respectively). The effective wetted rack lengths show that circular bars are less effective than T-shaped bars for the same gravel tests in the case of 30% slope, considered as the recommended design slope. Depending on the incoming flow to the rack, we can also conclude that lengths are near to the length proposed by Krochin [6] with $f$ = 30%, or near to the value of $f$ = 15% for higher incoming flows.

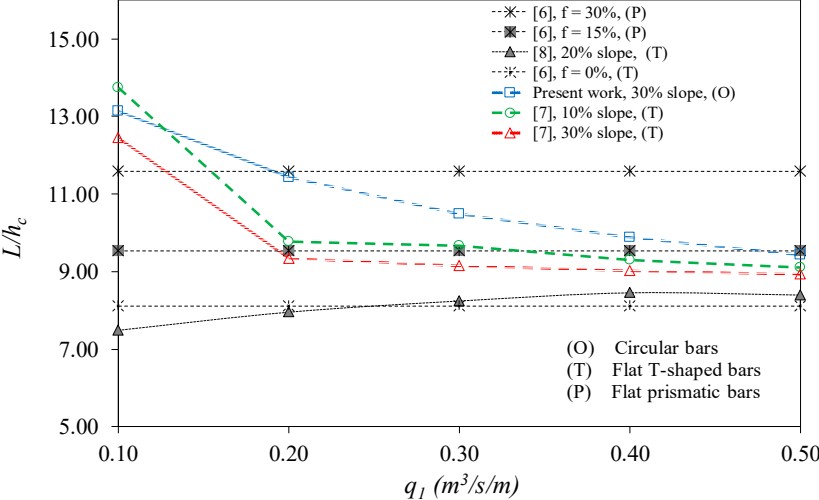

**Figure 20.** Calculated rack length necessary to derive the total incoming flow when sediment transport is presented for different rack slopes.

Table 6 summarizes the lengths calculated in Equations (6)–(10) and presented in Figure 20, in the case of T-shaped and circular bars for the initial $m$ = 0.28 and a longitudinal rack slope $tan\theta$ = 30%, as well as the case of T-shaped bars and $tan\theta$ = 10% slope.

**Table 6.** Summary of lengths calculated to obtain the effective wetted rack length considering gravel-sized sediments.

| Case | $q_1$ (m³/s/m) | $L_1m$ (m) | $L_1m'$ (m) | $Lm$ (m) | $Lm - L_1m$ (m) | $Lm - L_1m + L_1m'$ (m) | $Lm'/h_c$ (m) |
|---|---|---|---|---|---|---|---|
| T-shaped bars, 10% slope | 0.100 | 0.62 | 1.18 | 0.82 | 0.20 | 1.38 | 13.75 |
| | 0.200 | 1.02 | 1.32 | 1.26 | 0.24 | 1.56 | 9.78 |
| | 0.300 | 1.32 | 1.72 | 1.62 | 0.30 | 2.02 | 9.66 |
| | 0.400 | 1.64 | 2.06 | 1.94 | 0.30 | 2.36 | 9.29 |
| | 0.500 | 1.90 | 2.36 | 2.22 | 0.32 | 2.68 | 9.11 |
| T-shaped bars, 30% slope | 0.100 | 0.66 | 0.94 | 0.97 | 0.31 | 1.25 | 12.45 |
| | 0.200 | 1.08 | 1.08 | 1.49 | 0.41 | 1.49 | 9.34 |
| | 0.300 | 1.44 | 1.44 | 1.92 | 0.48 | 1.92 | 9.15 |
| | 0.400 | 1.76 | 1.76 | 2.29 | 0.53 | 2.29 | 9.02 |
| | 0.500 | 2.04 | 2.04 | 2.63 | 0.59 | 2.63 | 8.92 |
| Circular bars | 0.100 | 0.35 | 1.03 | 0.65 | 0.30 | 1.32 | 13.14 |
| | 0.200 | 0.56 | 1.40 | 0.99 | 0.43 | 1.83 | 11.44 |
| | 0.300 | 0.74 | 1.66 | 1.28 | 0.54 | 2.20 | 10.49 |
| | 0.400 | 0.895 | 1.88 | 1.52 | 0.63 | 2.51 | 9.89 |
| | 0.500 | 1.04 | 2.07 | 1.75 | 0.71 | 2.78 | 9.44 |

## 4. Conclusions

In Andean regions, bottom intake systems can constitute an adequate solution to provide water to communities located far from urban centers (paramos) where other types of intake systems that would require greater investments in operation and maintenance do not seem technically or economically feasible.

An important issue in the operation and maintenance of these intakes is the clogging of the racks due to solids that are trapped in the slits. This causes malfunction in bottom intakes and reduces the diverted flow. Design parameters to avoid this still need to be broadly studied and clarified; in this, the bar profile plays an important role. Moreover, not quite experimental works are presented in the bibliography to improve knowledge of and avoid this phenomenon.

Experimental campaign with 24 tests, wherein each test was repeated three times, with a flow transporting gravel-sized material were presented and the results were analyzed increase the understanding of the inefficiency of a circular bar profile in bottom intakes in comparison with T-shaped bars, as reported in previous works with the same sediment size. Although circular profiles have more efficiency in the case of clear water flow, in the case of sediment being transported, circular profiles tend to trap a wider range of diameters and the friction needed to drag gravels settled in the rack is higher. The rack slope does not considerably reduce the occlusion presented in circular bars.

The effective void ratio, *m'*, was calculated at the end of each test and was found to pass from the original 0.28 to values of 0.07–0.092 when the slope was increased. Images of the rack showed the important sediment trapped in the bars, which leads to an anomalous function of the intake and gives rise to more maintenance work. A model was proposed that allows the calculation of the wetted rack length, taking into account the gravel occlusion. Values were compared with commonly used formulations, such as those of Frank [8,9] and Krochin [6]. The lengths, taking into account gravel size and a 30% slope, were in the range of 15 to 30% of the occlusion factor proposed by Krochin [6].

Intensive experimental works with a wide range of profiles, void ratios, and gravel-sized sediment in incoming flows are needed to improve the knowledge of the influence of the bar profile in the trapping of gravel-sized materials in the slit between the bars which comprise the racks of bottom intake systems.

**Author Contributions:** J.T.G. planned the laboratory measurements, developed data treatment and analysis, and participated in the writing. L.G.C. carried out the analysis and application of the methodology, analyzed the results, and participated in the writing. P.L.H. carried out the laboratory measurements and data treatment and participated in the writing. J.M.C. planned and managed the laboratory measurements and the data analysis and participated in the writing. All four authors reviewed and contributed to the final manuscript.

**Funding:** The authors are grateful for the financial support received from the Seneca Foundation of the Región de Murcia (Spain) through the project "Optimización de los sistemas de captación de fondo para zonas semiáridas y caudales con alto contenido de sedimentos. Definición de los parámetros de diseño". Reference: 19490/PI/14.

**Acknowledgments:** The authors are grateful for the financial support received from the Seneca Foundation of the Región de Murcia (Spain) through the project "Optimización de los sistemas de captación de fondo para zonas semiáridas y caudales con alto contenido de sedimentos. Definición de los parámetros de diseño". Reference: 19490/PI/14.

**Conflicts of Interest:** The authors declare no conflict of interest.

## Notations

| | |
|---|---|
| $a, b$ | constant of adjustment depending on the shape of bars and the space between them in Equation (10) |
| $b_1$ | space between bars |
| $b_w$ | bar width |
| $C_D$ | drag coefficient of gravels |
| $\overline{C_{qH}}$ | mean discharge coefficient for energy head |
| $C_{qH}$ | discharge coefficient for flow depth |
| $C_{q0}$ | static discharge coefficient |

| | |
|---|---|
| $d_{50}, d_{90}, d_{10}, d_{min}, d_{max}, d_{50c}$ | characteristic diameters of gravels |
| $f$ | percentage of rack occluded |
| $g$ | gravitational acceleration |
| $H_0$ | energy head at the beginning of the rack in reference to the rack plane |
| $H_{min}$ | minimum energy head obtained when the Froude number equals the unity, and there is critical depth |
| $h_c$ | critical depth |
| $h_0$ | flow depth at the beginning of the rack |
| $k$ | obstruction parameter defined as $k = (1 - f)\, m$ |
| $Lm$ | wetted rack length |
| $Lm'$ | effective wetted rack length considering rack occlusion |
| $L_1 m$ | wetted rack length in the slit of two bars, considering the initial void ratio |
| $L_1 m'$ | wetted rack length in the slit of two bars, considering the effective void ratio |
| $m$ | void ratio |
| $m'$ | effective void ratio considering rack occlusion |
| $q_1, q_2$ | specific approaching and rejected flow, respectively |
| $U$ | mean velocity of the flow over the rack |
| $U_0$ | mean velocity of the flow at the beginning of the rack |
| $W$ | mean weight of the gravels deposited over the racks |
| $x$ | longitudinal coordinate along the rack |
| $v$ | kinematic viscosity of water |
| $\rho$ | density of water |
| $\theta$ | angle of the rack plane with the horizontal |
| $F_{D0}$ | drag force |
| $R_{e0}$ | Reynolds number calculated at the beginning of the rack |

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
