# Peer review of "Occlusion in Bottom Intakes with Circular Bars by Flow with Gravel-Sized Sediment. An Experimental Study"

_water, doi:10.3390/w10111699_

Reviewer 1 Report

The reviewer wants to thank the authors for their very interesting paper and the presented results for the influence of sediment on the efficiency of a bottom intake. He/she just want to add some small comments/questions/suggestions:

-      Title: the mentioning of the T-shaped bar type in the title could be misleading, hence it is only compared based on literature values and not a part of the investigation presented in this paper. The reviewer would suggest that the title is shortened and the fact of this comparison is included in the abstract. 

-      Abstract: if the values of the slopes and the discharge is mentioned than in the same detail (either range or boundary values) and please also include the information, that three repetitions were conducted. Otherwise it is not clear. 

-      General comment: It should be mentioned that there are principal two different approaches of the design of such a structure (at least in the Alps): (a) small spacing to prevent the entrance of gravel and also (b) a wider spacing, which only protects for big parts of the sediment. In this case the sediment is separated in a sand trap afterwards and only dimensions, which may cause a problem are restrained by the rack. If the sediment amount is too big, more sediment has to be flushed out and, in this case, more water gets lost … part of the optimisation process. Please just include this thought. 

-      Citation general: for example, in line (L) 60 the citations should be included in one [] also in L 62. Please check the whole paper. 

-      L96: m is not defined (only in Table 2). 

-      L102: to start with a subsection here is not good practice. Please introduce this instance after section 1 and maybe dived this section further. 

-      L137: the d50 is a good value but it would be further very interesting how much can theoretically fit through the bars and also the d90 … maybe also min and mix values. 

-      L150: solid weight includes a drying process previously or how is secured that different water content can bias the result.

-      Figure 4 and L161: please clarify which value is which … maybe by adding the variable to Fig 4

-      Section 3.1.2: L183 please explain this process with AutoCAD in detail and instead of the pictures an analysis of the covering of different sections would be very nice. Maybe split the rack in four sections and quantify the void ration for each one. The reviewer would assume that the first part is closed earlier or stronger … 

-      Fig. 11 -13: The comparison is very interesting but it should be addressed that in the higher discharges the curve is significantly higher at the trash rack start. How big is this effect? 

-      Fig 9 and Fig 20 are very similar and could be joint. 

-      Fig 21 the citation is not correct!

-      Fig 16- 19 the literature references and which parts are new or based on previous works should be always clear, without reading the text. Please clarify this. 

-      L369: “around” is not needed. 

-      L380: problem with citations. Check this please.

-      References: (1) the citations style is not consistent/ correct. (2) if the source is not in English the language should be mentioned (for example [18]) (3) typo in [13] Please check the references carefully. 

The reviewer wants to thank the authors ones more for their very infesting paper and is looking forward to read the corrected version. 

Author Response

The reviewer wants to thank the authors for their very interesting paper and the presented results for the influence of sediment on the efficiency of a bottom intake. He/she just want to add some small comments/questions/suggestions:

-      Title: the mentioning of the T-shaped bar type in the title could be misleading, hence it is only compared based on literature values and not a part of the investigation presented in this paper. The reviewer would suggest that the title is shortened and the fact of this comparison is included in the abstract.

Reply: Thank you for your suggestion. The title has now been shortened and the comparison with previous studies with T-shaped bars has been included in the abstract.

-      Abstract: if the values of the slopes and the discharge is mentioned than in the same detail (either range or boundary values) and please also include the information, that three repetitions were conducted. Otherwise it is not clear. 

Reply: Thank you for raising this. Following your ideas, the abstract has been modified.

-      General comment: It should be mentioned that there are principal two different approaches of the design of such a structure (at least in the Alps): (a) small spacing to prevent the entrance of gravel and also (b) a wider spacing, which only protects for big parts of the sediment. In this case the sediment is separated in a sand trap afterwards and only dimensions, which may cause a problem are restrained by the rack. If the sediment amount is too big, more sediment has to be flushed out and, in this case, more water gets lost … part of the optimisation process. Please just include this thought. 

Reply: Thank you for your comments. We really appreciate your suggestion and totally agree with you. This information has been included near the end of the Introduction section (lines 90-94).

-      Citation general: for example, in line (L) 60 the citations should be included in one [] also in L 62. Please check the whole paper. 

Reply: Thank you for raising this. These citations have been corrected. Following this idea, the entire manuscript has been checked.

-      L96: m is not defined (only in Table 2). 

Reply: Thank you for identifying this. The variable m represents the void ratio defined as the void area divided by total area of the rack. It is first described in line 86 of the current version.

-      L102: to start with a subsection here is not good practice. Please introduce this instance after section 1 and maybe dived this section further. 

Reply: We appreciate to the reviewer for his/her suggestion. The subsection 1.1 has been included in the Section 2. Experiment setting as 2.3. Previous studies of T-shaped Bottom Rack Occlusion by Flow with Gravel-Sized Sediment.

-      L137: the d50 is a good value but it would be further very interesting how much can theoretically fit through the bars and also the d90 … maybe also min and mix values. 

Reply: This is a very interesting commentary. According to the sieve curve of the gravels, on the coarse part of the sieve curve, d90 = 35 mm and dmax = 40 mm. On the finest part, the dmin = 10 mm, while the d10 = 16 mm. Considering the spacing between bars, b1 = 11.7 mm, very few materials will go through the rack for the tested gravel. This information has been included in the manuscript.

-      L150: solid weight includes a drying process previously or how is secured that different water content can bias the result.

Reply: Thank you for your question. Gravels are drained during a period of thirty minutes. After that, gravels were weighed. This information has been included at the end of Section 2.2 Sediment Experimental Tests with racks made of circular bars (m = 0.28).

-      Figure 4 and L161: please clarify which value is which … maybe by adding the variable to Fig 4

Reply: Thank you for your comment. The q1 is the incoming flow and q2 is the rejected flow measured in the V-notch weir located at the end of the rack. These parameters have been included in Figure 5.

-      Section 3.1.2: L183 please explain this process with AutoCAD in detail and instead of the pictures an analysis of the covering of different sections would be very nice. Maybe split the rack in four sections and quantify the void ration for each one. The reviewer would assume that the first part is closed earlier or stronger … 

Reply: Thank you for your comment. The reviewer is right about his/her assumption that lower void ratios are achieved at the beginning of the rack. From a top photograph of the rack, the occluded ratio is obtained and this is compared with the original void ratio. A new figure, Figure 7, that divides the rack in four parts has been included showing how the void ratio varies along the rack.

-      Fig. 11 -13: The comparison is very interesting but it should be addressed that in the higher discharges the curve is significantly higher at the trash rack start. How big is this effect? 

Reply: We appreciate so much this comment. Although the effect of clogging results in an increment of the flow depth along the rack in comparison with clear water flow, this effect is not observed at the beginning of the rack, where flow depths are in agreement. There was an error when considering the initial flow depth for the calculation of flow profile through Equations (2) and (3). This mistake has been corrected and Figures 11-13 are in agreement with observed data.

-      Fig 9 and Fig 20 are very similar and could be joint. 

Reply: We thank you for this comment. An attempt to do this has been carried out but these figures have too many information and it is more difficult the interpretation of their meaning. For this reason, figures were remained in the same way as in the previous version.

-      Fig 21 the citation is not correct!

Reply: Thank you for raising this. The citation in Fig. 21 has been corrected.

-      Fig 16- 19 the literature references and which parts are new or based on previous works should be always clear, without reading the text. Please clarify this. 

Reply: Thank you for indenting this in such a constructive way. The citation in the Figures 16-19 have been included. Besides this, the new data are plotted in red and the previous values in black.

-      L369: “around” is not needed. 

Reply: Thank you for this. The text has been changed following your suggestion.

-      L380: problem with citations. Check this please.

Reply: Thank you very much for this. Citations have been corrected.

-      References: (1) the citations style is not consistent/ correct. (2) if the source is not in English the language should be mentioned (for example [18]) (3) typo in [13] Please check the references carefully. 

Reply: Thank you for raising this. The reference list has been corrected.

The reviewer wants to thank the authors ones more for their very infesting paper and is looking forward to read the corrected version. 

Reply: Thank you very much for your encouraging words and ideas. Your suggestions have allow us to improve the manuscript.

Reviewer 2 Report

 The title, the methods and the language are generally appropriate for the aims of this publication.

The performed analyses are quite good and detailed and the overall quality of the paper is good, but some (few) issues need to be explained better. Moreover, the paper is quite long and it has a large number of figures; I suggest a reduction of both text and figures in order the paper to be more focused. I recommend a minor revision of the paper.

In particular, the following issues need to be addressed:
1) Line 130: how was the range of q1 chosen? What was the hydraulic regime (subcritical or supercritical) for such inflows and how did the hydraulic profile influence the occlusion phenomena and the diversion rate?
2) Line 137: be more specific instead of saying “almost uniform”
3) Line 138: Table “3”
4) Line 141: are the q1 the same of line 130?
5) Line 142: it should be more correct saying that 24 tests, with each test repeated 3 times, were conducted instead of 72, since you have combined 4 inflows with 6 slopes
6) Lines 148-150: how was the duration of the tests chosen? Was an equilibrium about the solid and liquid dynamics reached?
7) Eqs. 2-3: is the downstream boundary condition taken into account?

Author Response

The title, the methods and the language are generally appropriate for the aims of this publication.

The performed analyses are quite good and detailed and the overall quality of the paper is good, but some (few) issues need to be explained better. Moreover, the paper is quite long and it has a large number of figures; I suggest a reduction of both text and figures in order the paper to be more focused. I recommend a minor revision of the paper.

Reply: We want to thank the reviewer for his/her encouraging words and advice. The proposed comments have allowed us to improve the manuscript.

In particular, the following issues need to be addressed:

1) Line 130: how was the range of q1 chosen? What was the hydraulic regime (subcritical or supercritical) for such inflows and how did the hydraulic profile influence the occlusion phenomena and the diversion rate?

Reply: Thank you for your comments. To assure that the gravels are transported through the bottom, flows bigger than 0.070 m3/s/m are needed. With those considerations, the flow range was selected between 0.115 and 0.198 m3/s/m. The approximation flow regime is subcritical and changes to supercritical near the bottom racks. This information has been included at the end of the 2.1 Section.

The increment of incoming flow and flow profile produces an increment in the drag force exerted by the fluid on the embedded gravels that reduce the possibility of deposition of gravels on the slits of the bars. These ideas may be found in the next sections, in where some redaction changes have been done in order to clarify the occlusion effect.

2) Line 137: be more specific instead of saying “almost uniform”

Reply: We want to thank the reviewer for this/her comment. The text has been improved to include further details such as d10, dmin, dmax and d90 values of the sieve curve (please see Section 2.2)

3) Line 138: Table “3”

Reply: Thank you for raising this. The reference has been corrected.

4) Line 141: are the q1 the same of line 130?

Reply: Thank you for your question. In order to avoid misunderstandings, the flow ranges of the 2.1 Physical device Section were changed to the values used in this specific manuscript.

5) Line 142: it should be more correct saying that 24 tests, with each test repeated 3 times, were conducted instead of 72, since you have combined 4 inflows with 6 slopes

Reply: Thank you very much for your comment. Following your suggestion, the text has been changed.

6) Lines 148-150: how was the duration of the tests chosen? Was an equilibrium about the solid and liquid dynamics reached?

Reply: This is an interesting question. After all the sediment had passed over the rack or deposed in the spacing between bars, the tests duration was extended until no movement of the gravels was observed. During the dosage of gravels, It was achieved an equilibrium between supply and transport to avoid that all of the gravels passed through the rack at the same time. This gives rise to different duration that decrease with increment of incoming flow and slope of the rack, This information has been included at the end of section 2.2.

7) Eqs. 2-3: is the downstream boundary condition taken into account?

Reply: Thank you for your question. As the flow over the rack is supercritical in all the cases, the downstream boundary condition was not taken into account.

Reviewer 3 Report

The manuscript of Occlusion in Bottom Intakes with Circular Bars by Flow with Gravel-Sized Sediment. Experimental Study and Comparison with T-shaped bar-type is interesting.

I would like to reconsider decision after revisions.

I commented as follows;

1.(major)

The obtained knowledge in this study had lack versatility and generality.

The author should estimate and discuss dimensionless parameter (For example, Reynolds number and Capillary number).

2.(minor)

The present manuscript was relative long.

The author should explain the symbols and letters as Nomenclatures.

Author Response

The manuscript of Occlusion in Bottom Intakes with Circular Bars by Flow with Gravel-Sized Sediment. Experimental Study and Comparison with T-shaped bar-type is interesting.

I would like to reconsider decision after revisions.

I commented as follows;

 1.(major)

The obtained knowledge in this study had lack versatility and generality.

Reply: Many thanks for your comment.

Considering the current State of the Art of the bottom racks, most research papers take into account only clear water conditions and do not consider the influence of gravel occlusion. However, occlusion becomes the most important problem of these intakes. The Figure 3a shows the optimum profiles in case of clear water flows, where circular shapes are among the desirable bar types.

Through an experimental campaign, the current work shows that circular bars are not the most suitable with sediment transport, in contrast with the results obtained in the literature for clear water conditions.

The flow range of this study is from 0.115 to 0.198 m3/s/m, that covers part of the prototype flows, such as the Mazar – Dudas Hydroelectric Project (range from 0.100 to 0.500 m3/s/m). Besides, the void ratio of this study is of 0.28 with 0.03 m diameter of bars and 0.012 m spacing between bars (void ratio of 0.28). Those values are in the range of the proposed void ratios and spacing collected in Table 1 that summarizes the bibliography recommendations for field design of bottom intakes.

The sediment transport through bottom racks in the field cover a broad range. However, only sizes in the proximity of the spacing between two bars usually obstruct them. With this consideration, the gravel sieve curve was considered in this study.

As the flow range, the void ratio, the spacing between bars, and the gravel size tested are used in prototypes, the results obtained in this study could be considered of general application for design purposes.

With all the changed made during the reviewing process, we hope to have remarked the versatility and generality of the manuscript in a clearer way.

The author should estimate and discuss dimensionless parameter (For example, Reynolds number and Capillary number).

Reply: Thank you very much for raising this.

Some changes have been done in Section 3.2 of the manuscript.

Regarding the obstruction of the racks, the resultant force exerted by the fluid on the embedded gravels depend upon the geometrical form of the body, the roughness of its surface, its relative velocity, and the fluid density and viscosity. For a given geometrical form and roughness, the Buckingham Π theorem provides two dimensionless parameters, the Reynolds number and a dimensionless parameter related with the drag force. These variables together with the velocity, all measured at the beginning of the rack are analysed and presented in Fig. 14 of the manuscript, in relation with the ratio between the effective void ratio after obstruction divided by the original void ratio, m’/m. These variables are defined in Eqs. 4 and 5 as (please see file attached for these Equations):

where U0 is the velocity at the beginning of the rack; d50 the median diameter of the gravels; υ the kinematic viscosity of water; ρ the density of water; and CD the drag coefficient of gravels that adopts the value of 0.45.

In the figures, it can be observed that linear adjustments between m’/m ratio and the variables are achieved. Although a better correlation is found in case of the drag force, FD0, the velocity is proposed to be used to calculate m’/m ratio due that the velocity may be obtained in an easier way in practice (please see file attached for the figures).

Influence of other dimensionless number, like the Weber number, are found when analysing wetted rack length needed to completely derive an incoming flow. This analyses were taken in previous works for clear water experimental campaigns and can be reviewed in [23] reference.

 2.(minor)

The present manuscript was relative long.

Reply: We appreciate you bringing this to our attention. We have done several changes to try to reduce the length by omitting some figures of the manuscript.

The author should explain the symbols and letters as Nomenclatures.

Reply: Thank you for your suggestion. The nomenclature section has been included at the end of the manuscript.

Reviewer 4 Report

GENERAL APPRECIATION

The paper presents an experimental investigation of bottom intakes rack occlusion due to sediments, which is an important and up-to-date subject, interesting researchers and designers.

The research comprises experimental tests in bottom intakes with circular bars for different slopes and discharges. Authors compares the results with other tests made with different bar profiles and reach important conclusions. I found very interesting performance of circular bars comparing with different bar profiles. I consider that this manuscript has a very interesting approach and I agree with its publication in the journal.

Title is long but appropriate. In the abstract, subject is properly addressed and findings are properly highlighted. The manuscript is original and generally well structured (Introduction is badly numbered and results and discussion includes methodology) and well written (I just found one or two misprint). Figures and Tables are in a good quality. Introduction is badly numbered. It should not have 1 a lot of text and then 1.1. I would suggest taking 1.1 or include 1.1 Example of bottom intakes just after 1 or 1 and just a paragraph; 1.2 (line 56) Racks /Intake bar profiles 1.3 experimental test … . In alternative take off 1.1 and include it in 2. Experimental setting is short and Results and discussion includes some methodology. I suggest create a new iten with methodology, to include definition of the effective void ratio, m’and leave results and discussion to discussion of results. Lines 191/197-204 should be replaced, should not be placed in results and discussion.

SPECIFIC COMMENTS

Why not use image analysis to calculate m’?

Figure 10 shows for high flows some influence of slope (0.02).

Figure 21 should also present in a lower slope. Although circular bars doesn’t significantly change, with slope, T shape show slope influence.

DETAILED COMMENTS

Introduction is badly numbered: 1-Introduction; 1.1. Study of T-shaped Bottom Rack Occlusion by Flow with Gravel-Sized Sediment.

Line 56 needs a reference “Clogging of the racks is considered to be one of the most important causes of malfunction in bottom intakes.

Line 88 - Table 1 design parameters are general ?  related with circular bars ? If not there should be another column. Some parameters are different and should be reached in different condition, either bar profile or other. This should be mention.

Line 138 - this gravel is presented in Table 2. è Table 3

Figure 4 – An “A” is missing making it difficult to find section A-A

Flows and slopes (q1= 114.6, 138.3, 155.5, 198.0 l/s/m) and the six rack slopes (tanq = 0%, 10%, 20%, 30%, 33% and 35%) are to much repeated along the text

Figure 9 – Streamline is written in the Figure. However no streamline can be observed

Line 258 – Please change constant a because it is used in h0 .

Author Response

GENERAL APPRECIATION

The paper presents an experimental investigation of bottom intakes rack occlusion due to sediments, which is an important and up-to-date subject, interesting researchers and designers.

The research comprises experimental tests in bottom intakes with circular bars for different slopes and discharges. Authors compares the results with other tests made with different bar profiles and reach important conclusions. I found very interesting performance of circular bars comparing with different bar profiles. I consider that this manuscript has a very interesting approach and I agree with its publication in the journal.

Reply: Thank you very much for you encouraging words and advice. In turns, your comments have allowed us to improve the manuscript.

Title is long but appropriate. In the abstract, subject is properly addressed and findings are properly highlighted. The manuscript is original and generally well structured (Introduction is badly numbered and results and discussion includes methodology) and well written (I just found one or two misprint). Figures and Tables are in a good quality. Introduction is badly numbered. It should not have 1 a lot of text and then 1.1. I would suggest taking 1.1 or include 1.1 Example of bottom intakes just after 1 or 1 and just a paragraph; 1.2 (line 56) Racks /Intake bar profiles 1.3 experimental test … . In alternative take off 1.1 and include it in 2. Experimental setting is short and Results and discussion includes some methodology. I suggest create a new iten with methodology, to include definition of the effective void ratio, m’ and leave results and discussion to discussion of results. Lines 191/197-204 should be replaced, should not be placed in results and discussion.

Reply: Thank you for all your suggestions. Your ideas have been taken this into account to modify the manuscript. For instance, Section 1.1 is now located inside Section 2. Experimental setting. Besides this, a new subsection 2.4 Methodology to define the effective void ratio, m’ has been created. This new section includes the details of calculation of m’.

SPECIFIC COMMENTS

Why not use image analysis to calculate m’?

Reply: Thank you for your suggestion. In this study, we digitalized the photographs taken at the end of each test with AutoCAD. The images were used to draw the occluded areas. Due to the high range of pixel intensities registered in the images we preferred to use this method instead of using any image processing technique.

Figure 10 shows for high flows some influence of slope (0.02).

Reply: Thank you for raising this. This is an important observation. Although it was expected much more influence of the slope in the reduction of the occlusion process, the increment of the slope seems to reduce the obstruction of the racks. Some changes have been done in this section of the manuscript.

Figure 21 should also present in a lower slope. Although circular bars doesn’t significantly change, with slope, T shape show slope influence.

Reply: Thank for you for your suggestion. The case of T-shaped bars with 10% slope is now included in the Figure 21 and in the Table 6 and some comments are included in the manuscript in lines

DETAILED COMMENTS

Introduction is badly numbered: 1-Introduction; 1.1. Study of T-shaped Bottom Rack Occlusion by Flow with Gravel-Sized Sediment.

Reply: Thank you for bringing this into our attention. Numeration has been corrected in the manuscript.

Line 56 needs a reference “Clogging of the racks is considered to be one of the most important causes of malfunction in bottom intakes.”

Reply: Thank you for this. The reference has been included.

Line 88 - Table 1 design parameters are general ?  related with circular bars ? If not there should be another column. Some parameters are different and should be reached in different condition, either bar profile or other. This should be mention.

Reply: Thank you for your suggestion. This is a very interesting appreciation. A new column has been included in Table 1 with the profiles employed by each of the researches cited.

Line 138 - this gravel is presented in Table 2. è Table 3

Reply. Thank you for this. The reference has been changed. It should say Table 3.

Figure 4 – An “A” is missing making it difficult to find section A-A

Reply: Thank you for identifying this. Fig. 4 has been corrected.

Flows and slopes (q1= 114.6, 138.3, 155.5, 198.0 l/s/m) and the six rack slopes (tanq = 0%, 10%, 20%, 30%, 33% and 35%) are to much repeated along the text

Reply: We thank reviewer for this commentary it has been omitted in some parts of the manuscript.

Figure 9 – Streamline is written in the Figure. However no streamline can be observed

Reply: Thank you for this. Figure 4 has been corrected.

Line 258 – Please change constant a because it is used in h0.

Reply: Thank you for raising this. This constant “a” comes from coupling both previous equations. It is the same constant “a” presented the previous h0 equation.

Round  2

Reviewer 1 Report

The reviewer wants to thank the authors for their very valuable paper and the corrections. 

He/she would have introduced these two concepts earlier but OK. And I couldn’t find more details about the AutoCAD analysis, which would be very interesting.  The reviewer strongly disagrees with the comment to Figure 20. This could be easily joint with the Fig. 9 but it is just a repetition. 

Author Response

Reviewer 1:

 The reviewer wants to thank the authors for their very valuable paper and the corrections. 

 Reply: Thank you very much for your encouraging words and ideas. Your suggestions have allowed us to improve the manuscript.

 He/she would have introduced these two concepts earlier but OK. And I couldn’t find more details about the AutoCAD analysis, which would be very interesting.

 Reply: We thank you for your comment. The following details are now included in the manuscript (lines 215-222): Photographs of the top view of the occluded rack were taken at the end of each test to be processed later with a software in CAD design (AutoCAD). First, the photographs were imported and the size was adjusted to its real dimension. Then, a line was drawn over the gravels deposited along each one of the twelve slits of each rack. This line (in red colour) can be observed in Figures 7 and 8. Once the line was drawn its length was divided by the original length of the rack, that gives the percentage of reduction of the original void ratio. Next step is to calculate the average reduction between all the slits of the rack, that gives finally the effective void ratio. This is repeated for each of the tests taken.

 The reviewer strongly disagrees with the comment to Figure 20. This could be easily joint with the Fig. 9 but it is just a repetition.

 Reply: We appreciate to the reviewer for his/her suggestion. Old Figures 9 and 20 are now unified in new figure 5.

Reviewer 3 Report

The revisions was satisfied as my comments.

I recommended the acceptance for publication.

Author Response

Reviewer 3:

The revisions was satisfied as my comments.

I recommended the acceptance for publication.

Reply: Thank you very much for your encouraging words and ideas. Your suggestions have allowed us to improve the manuscript.